# A Comprehensive Map of FDA-Approved Pharmaceutical Products

**DOI:** 10.3390/pharmaceutics10040263

**Published:** 2018-12-06

**Authors:** Hao Zhong, Ging Chan, Yuanjia Hu, Hao Hu, Defang Ouyang

**Affiliations:** State Key Laboratory of Quality Research in Chinese Medicine, Institute of Chinese Medical Sciences (ICMS), University of Macau, Macau 999078, China; MB75822@umac.mo (H.Z.); YuanjiaHu@umac.mo (Y.H.); Haohu@umac.mo (H.H.)

**Keywords:** drug delivery system, FDA-approval drugs, generic drugs, oral sustained-release preparation, inhalation, complex injection, transdermal patch

## Abstract

With the increasing research and development (R&D) difficulty of new molecular entities (NMEs), novel drug delivery systems (DDSs) are attracting widespread attention. This review investigated the current distribution of Food and Drug Administration (FDA)-approved pharmaceutical products and evaluated the technical barrier for the entry of generic drugs and highlighted the success and failure of advanced drug delivery systems. According to the ratio of generic to new drugs and the four-quadrant classification scheme for evaluating the commercialization potential of DDSs, the results showed that the traditional dosage forms (e.g., conventional tablets, capsules and injections) with a lower technology barrier were easier to reproduce, while advanced drug delivery systems (e.g., inhalations and nanomedicines) with highly technical barriers had less competition and greater market potential. Our study provides a comprehensive insight into FDA-approved products and deep analysis of the technical barriers for advanced drug delivery systems. In the future, the R&D of new molecular entities may combine advanced delivery technologies to make drug candidates into more therapeutically effective formulations.

## 1. Introduction

During the pipeline of research and development (R&D) activities in the pharmaceutical industry, two key steps play an important role for revolutionary new drugs, including new molecular entities (NMEs) and novel dosage forms. In recent years, there is an obvious gap between growing productivity and R&D spending as the productivity of NMEs is constantly sluggish. The costs of NMEs are growing significantly at an average rate of 13.4% per year [1]. However, the success rate of NMEs in clinical trials is merely about 10% [2,3]. In 2017, only 46 NMEs were approved by the U.S. Food and Drug Administration (FDA) [4]. The high R&D costs and low NMEs output have pushed many pharmaceutical companies to advanced drug delivery systems. The R&D spending of new formulations are much lower than that of NMEs [5]. Moreover, current pharmaceutical products are far from optimal performance in clinical practice due to their low solubility, poor stability and poor targeting effect. Therefore, many novel dosage forms and drug delivery systems are developed to promote the clinical efficiency of drugs, reduce their toxicity and improve patient compliance. Furthermore, novel DDSs can greatly prolong the life cycle of NMEs. 

Pharmaceutics has experienced dramatic transformation over 60 years. Table 1 lists the milestones for drug delivery systems (DDSs) [6]. The 1st-generation (1950–1980) drug delivery systems (e.g., oral sustained release preparations, inhalations and transdermal patches) were developed rapidly, achieving high product translational efficiency [7,8,9]. In 1952, Spansule^®^ technology realized 12-h sustained release for the first time [10]. Four years later, the development of a pressurized metered dose inhaler (MDI) started the history of inhalation delivery systems [11]. In 1979, the approval of Scop^®^ made the transdermal patch available in the market [12]. When entering 2nd-generation, central issue focused on nanomedicines and smart delivery systems. However, although 2nd-generation (1980–2010) DDSs (e.g., liposomes, nanoparticle, microsphere and gene delivery) attracted lots of attention, there have been very limited products in the markets until now [7,8,9]. The liposome Doxil^®^ emerged as the first nanodrug in 1995 [13]. Abraxane^®^, the first nanotechnology-based target drug delivery, was available in the market in 2005 [14]. The commercial success of these DDSs always attracted a large number of followers. 

On the other hand, the release of the Drug Price Competition and Patent Term Restoration Act in 1984 (the Hatch Waxman Amendments) symbolized the beginning of the competition between brand and generic pharmaceutical companies [15,16]. After this, the abbreviated new drug application (ANDA), as a process for the review and approval of generic drugs products, has been established. This required generic manufacture to certify that generic is bioequivalent to brand drugs. During this period, generic drugs revealed a significant growing trend in the market, while brand drugs continued to be developed due to high benefits. Since the 1990s, a balance has been reached between new drug patent protection and generic drug entry [17]. Until now, the FDA has approved 34,673 drug products [18], including 114 administration routes [19] and 169 dosage forms [20]. There is no doubt that a product reference map will significantly promote our understanding of DDSs and dosage forms. However, there is no relevant research investigating the complicated distribution of pharmaceutical products. Drawing a reference map will be the first step to opening up access for subsequent drug delivery system studies at the product level.

The aims of this study were to provide a comprehensive map to FDA-approved pharmaceutical products. Firstly, the overall macro map of administration routes and dosage forms were plotted. Within this macro map, several advanced drug delivery systems were further analyzed, including oral controlled release formulations, inhalation delivery system, transdermal patch and complex injection formulations. Finally, the future perspective of pharmaceutics was discussed. 

## 2. Data Collection and Analysis

The pharmaceutical product data were compiled from the 38th edition FDA Orange book and drugs @FDA. The study included all “single” FDA-approved pharmaceutical products. “Single” for the purpose of the present analysis means that pharmaceutical products possess different approval numbers (shown as Figure 1). Data from the literature were extracted from the Science Citation Index Expanded (SCI-E) database via Web of Science (WOS), with the keyword searching strategy within the 37-year period from 1 January 1980 to 31 December 2017. All clinical data came from the largest clinical trial registration website in the world (https://clinicaltrials.gov/).

With the aim of investigating the current distribution of pharmaceutical products, only marketed pharmaceutical products were considered. Firstly, the overall macro map was plotted by the classifications of formulations. The administration routes were divided into six classifications labeled as oral, injection, inhalation, mucosal, cutaneous and others. To further analyze the distribution, dosage forms were mapped based on administration routes. In this case, one drug containing multiple formulations was assigned to each of the classes simultaneously. 

With this map, we further explored the development footprint of advanced drug delivery systems chronologically, including oral sustained release preparations, inhalation, transdermal patch and complex injection delivery systems. The analysis mainly included two parts for each DDSs; the translational efficiency and features of DDSs. 

## 3. The Overview of Pharmaceutical Products

### 3.1. Outline of Annual Approval and Discontinued Number

A total of 34,673 drug products have been approved up to 2017. Figure 2a shows the timeline of annual drug approvals after 1981 because the Orange Book did not show the approval date before 1981. New drug approvals (including new molecular entity, active ingredient, dosage form, combination, formulation and indication) reveal a relatively stable trend during this period. The subtle peak of new drug approvals emerges in 1996, which may be associated with an application backlog by the release of prescription drug user fee act in 1992 [30]. Moreover, the approval of biopharmaceutical drugs could also be viewed in the Figure 2a. Since the first biopharmaceutical drugs (Humulin R^®^) was approved by FDA in 1982, it is obvious that the number of biopharmaceutical products show a considerable growth. In 2017, new biological entities (NBEs) approved by the Center for Biologics Evaluation and Research (CBER) have occupied nearly one quarter of the new molecular entities, which significantly meet unmet clinical needs. Moreover, gene and cell therapy are attracting widespread attention. For example, the first gene therapy Kymriah^®^ was approved by the FDA [31]. 

Annual generic drug product approvals indicated the significant increase trend. Since 1982, the curve showed a rising trend. In 1988, the curve reached the peak, four years after the enactment of the Hatch Waxman Amendments. However, the number of generic drug approvals dropped sharply from 1988 to 1990. After the 1990s, the number of approved drugs returned to a stable level. After 2014, generic drug products witnessed significant growth, which may be related to the rising difficulty of new drug R&D and control of drug costs. In addition, the patent expiry of biologics has also opened the door to the so-called biosimilars [32]. In 2015, the first biosimilar (Zarxio^®^) was approved by the FDA. The biosimilar market keeps growing due to patent expiration of several important biologics. As of 2017, a total of 10 biosimilars have been approved. What’s more, the FDA released the Biosimilar Action Plan in 28 July 2018 to help biosimilar development more efficiently.

Figure 2b,c displayed annual marketing and discontinued a number of new drugs and generic products by 2017, respectively. It is obvious that new drugs enjoy a longer life cycle than generic drugs because the generic drugs in the early stage are more likely to be withdrawn from the market. These discontinued drugs were recalled from the market due to different reasons ranging from safety (e.g., adverse side effects etc.), lack of efficiency, manufacture issues, regulatory changes to the financial burden [33].

### 3.2. Proportion of Route of Administration

The distribution of administration route of current marketed pharmaceutical products is shown in Figure 3. Overall, the oral delivery route (62.02%) makes the largest contribution to pharmaceutical products, followed by injection (22.43%), cutaneous (8.70%), mucosal (5.22%), inhalation (1.21%) and others (0.42%) (Figure 3a). The results reveal that oral delivery remains the most appealing route due to high patient compliance and ease of administration. Generic drug companies are more likely to develop traditional administration routes in comparison to new drugs. For example, the proportion of oral generic products have far exceeded 60% (Figure 3b).

The ratio of generic drugs to new drugs is used to evaluate the level of technical barriers and market capacity. Generally speaking, the lower the technical barriers and the greater the market capacity, the higher the ratio of the generic drug to a new drug. Table 2 clearly shows the ratio value of different administration routes (oral: 4.69 > injection: 2.30> cutaneous: 2.03 > mucosal: 1.61 > inhalation: 0.97). The highest ratio of oral delivery system symbolizes the lowest technical barrier. So oral brand-name drugs are more likely to struggle with generic drugs entry after the patents expired. Conversely, inhalation delivery systems with low ratio show the high technical barrier. New inhalation drugs may enjoy a longer period of market exclusivity, due to the high technical barrier to the entry of generic manufactories. 

### 3.3. Distribution of Formulations

Figure 4 shows the distribution of dosage forms by administration routes. Overall, oral administration reveals more flexibility in dosage form design than other delivery routes. While inhalation administration shows certain limitations, it relates to complex formulations and devices. The formulations of new drugs are more diverse than that of generic drugs. Generic drugs mainly concentrate on conventional dosage forms, while new drugs are more likely to develop novel formulation. 

**Oral formulations:** Oral administration has been the first choice of DDSs when a new drug is developed because of its easiness of administration and high acceptance by patients. If a drug was taken orally, the drug must firstly be released from the formulation, dissolved in the intestinal fluid and passed through the gastrointestinal membrane. Oral administration reveals more flexibility in formulation design than other administration routes, which comprises nearly all dosage forms ranging from solution, suspension, emulsion, powder, granule, capsule to tablets and so forth. The high ratio of generic drugs to new drugs reveals that oral formulations have developed into the mature stage. There are a large number of generic drugs available for almost all oral formulations. For example, the number of generic oral sustained release preparations is nearly four times that of new products. However, several new formulations (e.g., oral liquid Ravicti^TM^ [34], oral soluble film Zuplenz^®^ [35,36] and effervescent tablet Binosto^TM^ [37]) for swallowing difficulties have no generic competitors due to the patent protection, which indicated the new formulation patents are also important tools to extend the life cycle of drugs. For example, Zuplenz^®^ is a unique formulation of ondansetron developed using PharmFilm^®^ technology. This technology has been granted a U.S patent in 2010, which are providing intellectual property protection for the company’s film products and methods of their preparation. Thus, Zuplenz^®^ could enjoy a long-term market exclusivity.

**Injection formulations:** Injection formulations, which are mainly made into liquid and powder state, are able to let drugs directly access to the bloodstream for rapid onset, even targeting specific organ and tissue sites. The analysis in 3.2 reveals that injection administration is relatively easy to reproduce. For example, conventional injectable formulations (e.g., injection solution and injectable powder) have accounted for over 90% of total injection formulations. Moreover, the number of generic products is over three times than these simple new formulations. However, there are few generic products for complex injection formulations (e.g., liposome, emulsion, suspension, microsphere, nanoparticle and implant) because these complex formulations hold very high technical barrier. We will perform a deep analysis on complex formulations in Section 4.4.

**Cutaneous formulations:** Cutaneous administration delivers the drugs across the skin barrier for topical effect or the systemic circulation. Cutaneous formulations are designed to be suitable for external use. The majority of cutaneous administration drugs are conventional dosage forms (e.g., cream, solution, ointment and lotion), which are easily reproduced by generic manufacturers. However, there are only 54 transdermal patches (28 new products and 26 generic products) in the market now, which indicate that transdermal patches have relatively high technical barriers. 

**Mucosal formulations:** Mucosal administration route (e.g., nasal, buccal, ophthalmic, vaginal, rectal, sublingual and intrauterine) provides many benefits over other administration routes, such as noninvasive administration, rapid onset and elimination of hepatic first-pass metabolism. Current marketed mucosal dosage forms are mainly solution, spray, tablet, ointment, cream and chewing gum, which accumulated 84% among the total amount. These products are relatively easy to be reproduced by generic manufacturers. 

**Inhalation formulations:** Inhalation formulations show a rapid and predictable onset of action [38]. Overall, inhalation administration is the most difficult to be reproduced comparing with other administration routes. Figure 4 shows that generic drugs mainly concentrate in liquid state formulations (solution, suspension and liquid). In other word, new drugs based on these formulations are more likely to struggle with the entry of generic followers. Aerosol, powder and spay inhalation drugs have very few generic products and enjoy a period of market exclusivity, due to the high technical barriers of the drug/device combination. 

## 4. The Analysis of Advanced DDSs

The above analysis clearly revealed that 1st generation of DDSs (1950s–1970s) with numerous products in clinic reached the mature stages, while 2nd generation of DDSs (1980s–2010s) with a few successful products existed high technical barrier. These advanced drug delivery systems (DDSs) are widely studied in both academia and the pharmaceutical industry [39]. There has been a dramatic increase in the publication number and clinical trials on novel DDSs during the past three decades [40]. However, an obvious gap can be viewed between product output and research input. These advanced pharmaceutical technologies with high technical barrier were necessary to further investigate and analyzed the translational efficiency and clinical success rate. Therefore, several widely-used DDSs with high technical barrier were collected, including oral sustained release preparations, transdermal patch, inhalation delivery (aerosol, powder and spray) and complex injection formulations (liposome, emulsion, microsphere, nanoparticle and nanosuspension).

In Figure 5, 10 DDSs were classified into 4 types according to two parameters (the ratio between global clinical trials and global publications and the ratio of FDA approved products to clinical trials in the US) (the ratios were calculated by data in Table 3). The horizontal axis represents the translational efficiency—the ratio between global clinical trials and global publications. The vertical axis shows the cumulative clinical success rate—the ratio of FDA approved products to clinical trials in the US. Five percent of the horizontal value and 10% of the vertical value were chosen as the original points of the four quadrants and the taxonomy for these drug delivery technologies. A ratio of 5% between global clinical trials and global publications represents a key point of translational efficiency from academic research to clinical trials, while a 10% clinical success rate is the average clinical success rate of NMEs in the past 20 years [2].These advanced drug delivery technologies can be classified into four quadrants in a coordinate system. The first type represents technologies with a high translational efficiency and high clinical success rate, comprising transdermal patch and oral sustained release preparation. The second type has a low translational efficiency and high clinical success rate and there is no DDS belonging to this category. The third type represents high translational efficiency and low clinical success rate, including inhalation (aerosol, powder and spray) and complex injections (emulsion, liposome, nanoparticle, suspension and microsphere). The fourth type has very low translational efficiency and low clinical success, such as nanoparticle. 

### 4.1. Oral Sustained Release Preparations

Oral sustained release formulations are able to control drug release at a predetermined rate to achieve a prolonged therapeutic effect, maintaining desired drug concentration either in blood plasma or at target site [41]. 

The first oral sustained drug (Dexedrine^®^) in 1952 by Smith Kline & French achieved 12-h drug release at the first time. Compared with conventional dosage forms, Dexedrine^®^ has unique benefits, including the reduction of the frequency of dosing, improvement in patient compliance and less side effects. Commercial success of Dexedrine^®^ promoted further development of sustained release technologies. After that, four different drug release mechanisms were gradually established to accelerate the development of sustained release formulations, including dissolution-controlled, diffusion-controlled, osmosis-based and ion-exchange mechanisms [41]. Several novel oral sustained release technologies entered into market. Procardia XL^®^ based on osmotic-controlled release oral delivery system technology was approved in 1989 [42]. Osmotic pump technology can achieve different drug release behaviors by adjusting the osmotic pressure difference between environment and the drug system [42]. Sular^®^ based on Geomatrix^TM^ technology was marketed in 1995. This kind of tablets consisted of drugs-containing layer and retardation layers. Drugs-containing layer was coated by retardation layers on both sides. The drug release was dependent on the proportion of drugs-containing and retardation layers [43]. In 2005, Glumetza^®^ based on Acuform^TM^ technology could make active ingredients be absorbed in the upper part of the small intestine with a sustained release effect [44].

Until 2017, total 192 oral sustained release preparations are available in the market. These drugs mainly were made into three dosage forms comprising tablet, capsule and suspension. As shown in Figure 5a, oral sustained release tablet occupied the largest share with 61%, followed by capsule (34%) and suspension (5%). Figure 6b reveals that oral sustained release preparations show a stable increasing trend in the market.

### 4.2. Inhalation Systems

Lung with an oak-tree like structure has many unique properties, such as huge surface area (100 m^2^), rich blood capillary and relatively few metabolic reactions. Thus, pulmonary drug delivery performs a great capability for both systemic and localized drug delivery [45,46]. At present, there are three kinds of inhalers in the market, including metered dose inhaler (MDI), dry powder inhaler (DPI) and nebulizer [47]. 

First Inhaled drug with MDI was approved in 1956 by 3M company for treating bronchial asthma [48]. MDI is made up of propellants, drugs and additives (e.g., solvent, emulsifier et al.), offering an excellent ability to control the therapeutic dose via valve system. Due to the ozonosphere destroy, Freon (CCl_3_F, CCl_2_F_2_, CClF_2_-CClF_2_) as propellants was prohibited gradually. Two hydrofluoroalkanes (HFA134a, HFA227) were approved by the FDA in 1994 to replace Freon [49]. At the same time, Norisodrine^®^ with DPI was developed in 1971 [50]. For dry powder inhaler, drug particle size should be controlled at specific sizes. Drugs and carriers are stored in capsule, blisters and multiple doses form. In comparison to other pulmonary dosage forms, DPI has some benefits, including the easily-portable administration, less cleaning steps and low contamination risk [51]. However, the drug release and absorption of DPIs are strongly dependent on the inhalation by patients.

Many drugs have been developed for the treatment of asthma and chronic obstructive pulmonary disease (COPD). Today, over 60 inhalation products (aerosol, powder and spray) are available in the market. Among these drugs, powders for inhalation account for 55%, followed by aerosols with 36% and 9% of sprays. It is obvious that the aerosol number shows a decrease since 2011, which relates to the discontinuation of chlorofluorocarbon inhalers [52]. On the other hand, powder inhalations rise rapidly between 2012 and 2017, which shows a great potential prospect. 

Lung also has specific advantages on peptide or protein delivery, such as rapid absorption and less interference from proteases [53]. Exubera^®^ (Pfizer) and Afrezza^®^ (Mannkind), as the insulin inhalation, were approved in 2006 and 2014 respectively [54,55]. However, these two products failed to reach commercial success due to their low bioavailability and patient compliance. 

### 4.3. Transdermal Patch

The structure of skin is composed of cuticle, epidermis and dermis [56]. In general, transdermal patch delivers drug via cuticle to dermis where distributes the blood vessels and drugs are absorbed into the systemic circulation at here. 

Transdermal patches mainly comprised three layers, including a closed backing layer to prevent drugs loss, drug-containing layer to store drugs and adhesive layer to keep the patched in contact with skin [57]. Transderm Scop^®^ by Glaxosmithkline was the first FDA-approved patch, which can deliver scopolamine to prevent nausea and vomiting from motion sickness for up to 3 days (72 h) [12]. After the entry of Transderm Scop^®^, an increasing number of APIs were developed into patches. According to pharmacokinetics principles, current transdermal patches could be divided into three types, including adhesive dispersion, matrix-diffusion and reservoir [34] 

Until 2017, total 28 new transdermal patches still exist in the market. As shown in Figure 6c, 72% of patches is drug-in-adhesive type, while drug-in-matrix type is 16% share and drug-in-reservoir type has only 12%. All marketed reservoir patches were approved before 1999 because this type of patches leads to uncontrolled drug release from the reservoir. 

### 4.4. Complex Injections

From the above analysis, it is obvious that complex injections have low translational efficiency and clinical success rate. In this part, four complex injection preparations are further discussed, including liposome, emulsion, nanoparticle and suspension.

#### 4.4.1. Liposome Injections

Liposome is an artificial membrane, which mainly is prepared by either natural or synthetic phospholipids (e.g., DLPC DMPC DPPC et al.). The drugs are encapsulated in the vesicle. With the liposome gradually degrades, drugs were released slowly [58]. 

The first FDA-approved liposome Doxil^®^ in 1995 with doxorubicin hydrochloride is used to treat ovarian cancer, AIDS-related Kaposi’s Sarcoma and multiple myeloma [13]. Compared with free doxorubicin, Doxil^®^ revealed greater efficacy and lower cardiotoxicity benefiting from long circulation half-life (45 h) and passive targeting property. Moreover, Doxil^®^ could also targets the tumors by the enhanced permeability and retention (EPR) effect.

Doxil^®^ achieved great commercial success with up to $600 million sale peak in 2001 [59]. Thus, many pharmaceutical manufactories entered into liposomal area. Ambisome^®^ was approved in 1997 for treating deep fungal infections [60] and Depodur^®^ (2004) [61] and Exparel^®^ (2011) [62] entered into the market as an anesthetic. Depocyt^®^ (1999) [63], Marqibo kit^®^ (2012) [64] and Onivyde^®^ (2015) [65] were approved for cancer therapy. Table 4 summarized that total ten drugs were approved by FDA, including eight new liposome drugs and two generic products. These two generic drugs were therapeutic equivalents with Doxil^®^, which lead to continuous decline in sales of Doxil^®^. The success of Sun pharma and Dr Reddys Labs revealed a great opportunity for generic liposome products.

#### 4.4.2. Nanoparticle and Suspension Injections

Nanoparticle is a microscopic particle with at least one dimension less than 100 nm Since 1990s, nanoparticle drug delivery systems attracted widespread attention in pharmaceutical researches. 

Abraxane^®^ [14], an albumin-bound nanoparticle formulation of paclitaxel, achieved great commercial success for the treatment of recurrent breast cancer. In the formulation of Abraxane^®^, Paclitaxel has significant advantages over pure paclitaxel. Moreover, Abraxane^®^ does not contain phospholipids, which can avoid the hemolysis reaction of liposome [66]. Nowadays, total 13 suspension and nanoparticle injection products are available in the market. Among these products, several drugs with novel technologies attract widespread attention. Invega Trinza^®^ [67] with nanocrystal technology releases slowly over a long period of time. Sublocade^®^ [68] with the Atrigel technology in situ gel forming system is thought to be an epoch-making product.

#### 4.4.3. Microsphere Injections

Microspheres are small spherical microparticles, with diameters typically ranging from 1 μm to 300 μm. The key to develop the sustained-release injectable microspheres is how to choose an appropriate biodegradable polymer. Current marketed products mainly use synthetic polymers. For example, as synthetic polymers, PLGA and PLA are both biodegradable and biocompatible [69]. 

The first marketed product (Lupron depot) of sustained-release microsphere in U.S. was developed in 1985 by Takeda. Lupron depot [70] containing a gonadotropin releasing hormone (GnRH) agonist is used for the palliative treatment of advanced prostatic cancer. Currently, 11 FDA-approved sustained release microsphere injections are available in the market (listed in Table 4). For example, Sandostatin Lar^®^ Depot [71] solved the short biological half-life of peptide drugs, extending the dosing period to 4 weeks and improving patient compliance. Risperdal Consta^®^ [72] conquered the difficulty of medication and the abuse of drugs for the mentally ill.

#### 4.4.4. Emulsion Injections

Emulsion injection is a colloid system of two or more immiscible oil and water. The stability of emulsion depends on the interfacial tension between the oil and aqueous phase. 

Intralipid^®^ [73] is the first FDA-approved injectable emulsion in 1975, which was used as a source of calories and essential fatty acids after intravenous administration. Intralipid^®^ is a soybean oil-in-water formulation, stabilized by the egg phospholipid emulsifier. Multiple advantages of emulsions could be viewed, such as solubilization, buffering, passive targeting and improving stability.

## 5. Future Perspective

Drug delivery systems play a more and more important role in pharmaceutical R&D. Advanced drug delivery systems not only promote the efficiency of drugs but also extend the life cycle of NMEs. Our research provided a quantitative analysis of FDA approved products and discussed the technical barriers of DDSs according to two parameters. 1st-generation DDSs had achieved big success in the market, while 2nd-generation DDSs (liposome, nanomedicine and microsphere et al.) showed less success despite huge publications and funding over the past three decades. How to build the bridge for the gap from basic pharmaceutical research to the clinic is important for pharmaceutical scientists.

With the development of computer capability and algorithms, one new field “computational pharmaceutics” is emerging [75], which integrates artificial intelligence, big data and multi-scale modeling in drug delivery for in silico formulation design. Current pharmaceutical formulation development mainly depends on an experimental trial-and-error pattern, which is time- and money-consuming. With the accumulation of a large amount of experimental data, the R&D of pharmaceutical formulations will transform from trial-and-error experiments to data-driven AI. Combining with artificial intelligence algorithms, the quantitative model is well established for formulation prediction [76]. For example, the prediction model of oral disintegrating tablet (ODT) formulations was established using neural network techniques [77]. In this study, a total of 145 formulations were obtained from published papers in the web of science, which were used to establish the prediction model. The results showed that the accuracy of the DNN reached 80%. However, as R&D of drugs becomes more difficult, the traditional R&D pattern has great challenges. So how to make use of the data becomes critical for the future development of pharmaceutical formulations. Furthermore, molecular modeling approaches also become an important tool for formulation design. Interaction between drugs and excipients and related parameters could be calculated to evaluate system stability and combine strength, which guide drug prescription screening and obtain the best prescription quickly. Zhao et al.’s study successfully combined molecular modeling and experimental methods to get better lutein–cyclodextrin multiple-component formulations [78]. 

Smart materials for drug delivery (e.g., biomimetic materials) are also another important area. For example, liposome simulates the cell membrane for drug delivery. Human albumin is used in nanoparticle formulations (e.g., Abraxane^®^). Virus-like particles or cells are also important carriers for gene therapy. The first gene therapy named Kymriah^®^ was approved by the FDA in 2017, which is made from the patient’s own white blood cells and is a prescription cancer treatment used in patients who have acute lymphoblastic leukemia (ALL) [79]. How to design clinically-oriented new materials also plays an increasingly important role in drug delivery. However, although much research into material design for drug delivery has been published in the past 30 years, very few products were applied clinically. 

On the other hand, novel drug/device combination is also a smart method for drug delivery. In recent years, micro devices attracted increasing attention, which range from simple tongue depressors to micro-chip technology. One good example is inhalers, which are developed for solution inhalation. Moreover, 3D printing technology has found its way into the flexible manufacture of pharmaceutical products. The first oral disintegrating medicine Spritam^®^ made using 3D printing technology for antiepileptic medicine was approved in 2015 [80]. Digital health has been used to better manage and track health for patients. For example, Abilify mycite^®^ with an ingestible event marker sensor was approved by the FDA in 2017. When patients have ingested drugs, their doctors can easily track the signals [81]. 

In the future, the R&D of new molecular entities may combine advanced delivery technologies to make drug candidates into more therapeutically effective formulations.

## Figures and Tables

**Figure 1 pharmaceutics-10-00263-f001:**
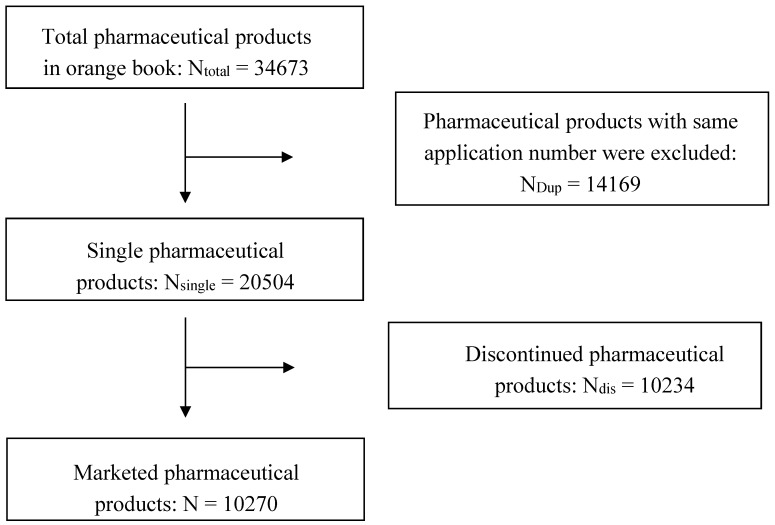
Marketed pharmaceutical products flow chart.

**Figure 2 pharmaceutics-10-00263-f002:**
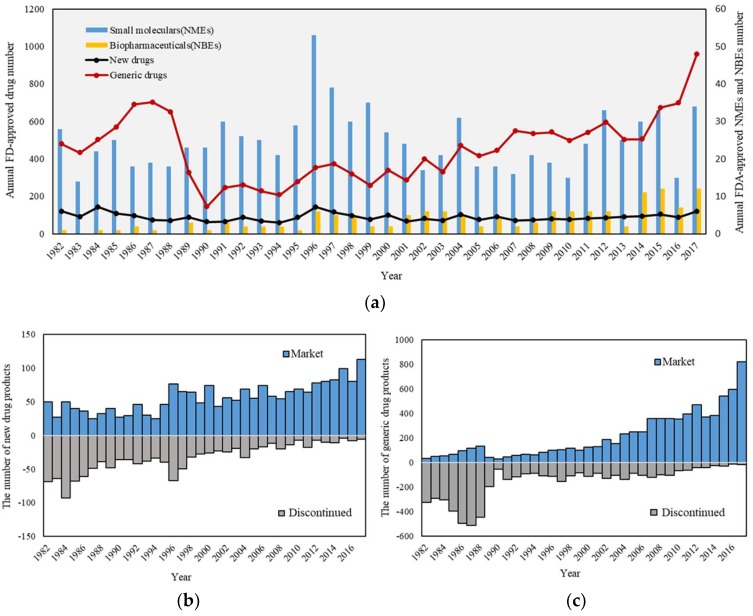
(**a**) The timeline of FDA-approved pharmaceutical products. (**b**) The number of marketed and discontinued new drugs. (**c**) The number of marketed and discontinued generic drugs. The blue columns represent marketed drugs and the grey columns represent discontinued drugs.

**Figure 3 pharmaceutics-10-00263-f003:**
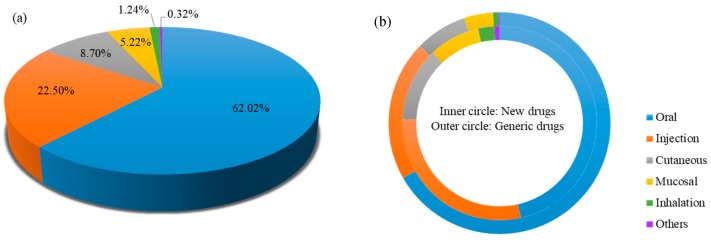
(**a**) The overall distribution of administration route of FDA-approval pharmaceutical products. (**b**) The proportion of administration route. Inner circle represents drugs; Outer circle represents generic drugs.

**Figure 4 pharmaceutics-10-00263-f004:**
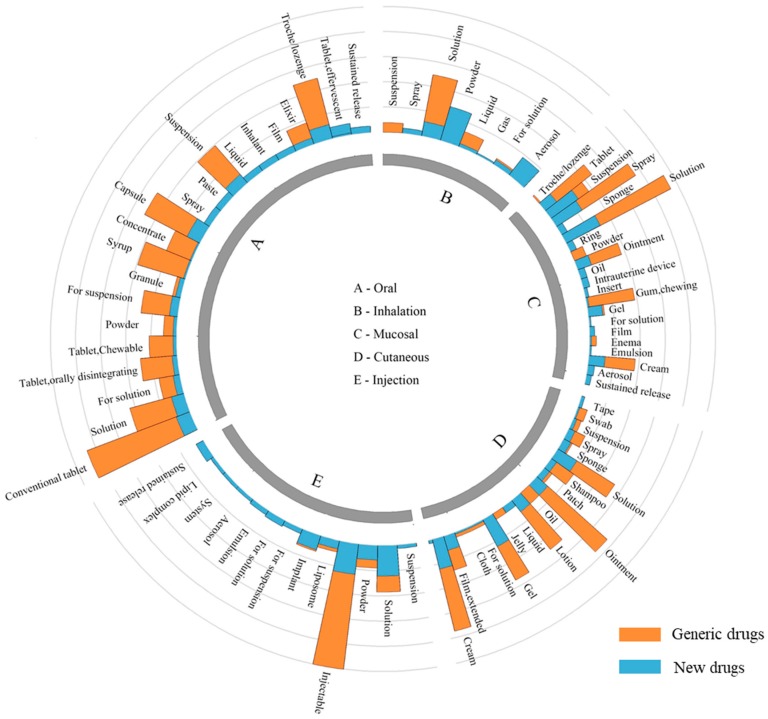
The distribution of dosage forms for each administration route. The orange column represents the number of generic drugs, the blue column represents the number of new drugs.

**Figure 5 pharmaceutics-10-00263-f005:**
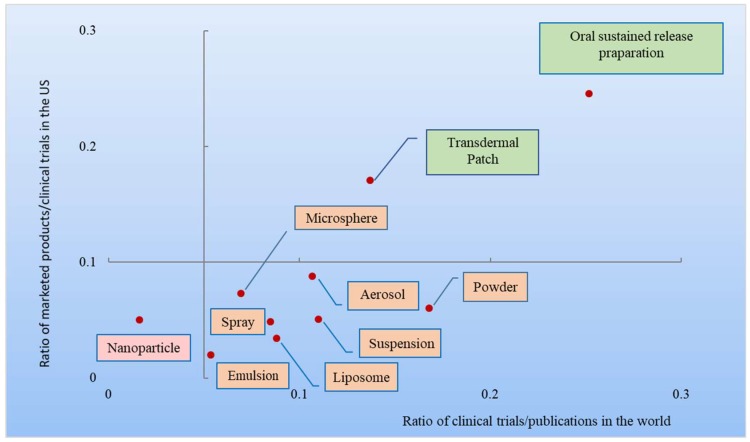
Four types of the 10 advanced pharmaceutical technologies.

**Figure 6 pharmaceutics-10-00263-f006:**
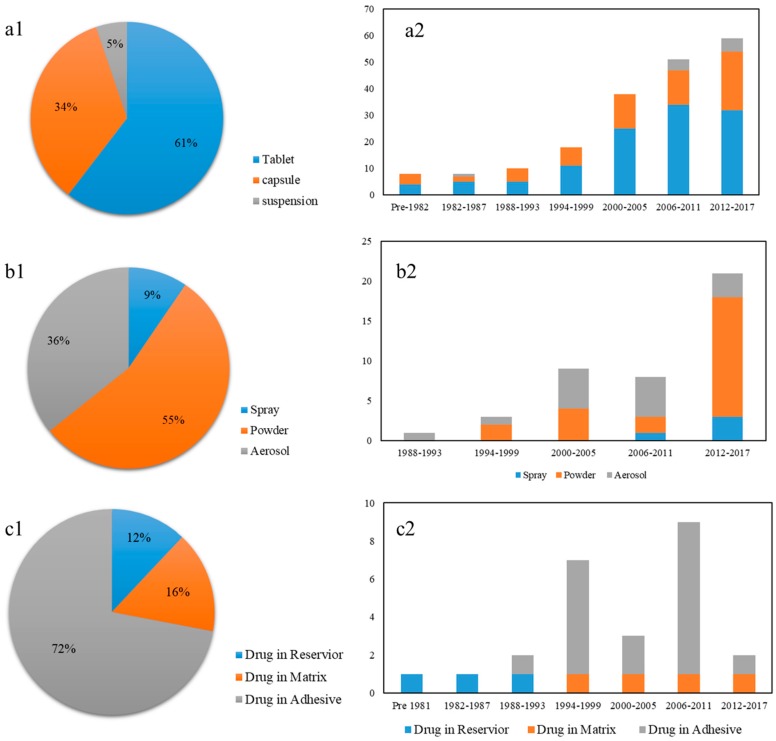
(**a1**) The distribution of dosage forms of marketed new oral sustained release drugs; (**a2**) Evolution of oral sustained release preparations. (**b1**) The distribution of dosage forms of marketed new inhalation drugs: spray; powder; aerosol; (**b2**) Evolution of new inhalation drugs. (**c1**) The distribution of dosage forms of marketed new transdermal patches: drug in reservoir, drug in matrix and drug in adhesive; (**c2**) Evolution of new transdermal patches.

**Table 1 pharmaceutics-10-00263-t001:** The landmark of key drugs delivery technologies to the market.

Year	Drug Delivery System	
1952	The first sustained-release technology Spansule^®^	[10]
1956	The first pressurized metered dose inhaler (MDI)	
1969	The first dry powder inhalation (DPI)	[11]
1979	The first transdermal patch Transdermal Scop^®^	[21]
1982	The first recombinant human insulin Humulin R^®^	[22]
1984	The first Biodegradable microsphere Vivitrol^®^	[23]
1986	The first injection microsphere Decapeptyl^®^	[24]
1989	The first Push-Pull Osmotic Pump product Procardia XL^®^	[25]
1995	The first FDA-approved liposome Doxil^®^	[13]
2005	The first FDA-approved nanoparticle Abraxane^®^	[14]
2006	The first FDA-approved botanical medicine Veregen^TM^	[26]
2015	The first FDA-approved 3D print drug Spritam^®^	[27]
2017	The first FDA-approved gene therapy Kymriah^®^	[28]
2017	The first FDA-approved digital drug Abilify MyCite^®^	[29]

**Table 2 pharmaceutics-10-00263-t002:** The ratio of generic drug quantity to new drug quantity.

Route	Number (New Drugs)	Number (Generic Drugs)	Ratio	Ranking
Oral	1119	5252	4.69	1
Injection	702	1609	2.29	2
Cutaneous	295	599	2.03	3
Mucosal	205	331	1.61	4
Inhalation	66	61	0.92	5

**Table 3 pharmaceutics-10-00263-t003:** Numbers of publications, clinical trials and marketed products of 10 advanced pharmaceutical technologies (1980–2017).

Key Drug Delivery Technologies	Number of Global Publications	Number of Global Clinical Trials	Ratio of Clinical Trials to Publications (%)	Number of Clinical Trials in US	Number of Marketed Products in the US	Ratio of Products to Clinical Trials in the US (%)
Oral sustained release preparations	7150	1798	25.15	859	205	24.55
Transdermal patch	4161	570	13.70	323	54	17.09
Aerosol inhalation *	3204	342	10.67	171	16	9.36
Powder inhalation *	5227	878	16.80	383	25	6.53
Spray inhalation *	2284	194	8.49	82	4	4.88
Liposome injection ^#^	3885	342	8.80	232	8	3.45
Emulsion injection ^#^	5033	269	5.34	119	5	4.20
Microsphere injection ^#^	1329	92	6.92	45	11	7.32
Suspension injection ^#^	3558	58	1.63	40	2	5.00
Nanoparticle injection ^#^	5468	601	10.99	276	11	5.07

* belong to inhalation delivery system. ^#^ belong to injection delivery system.

**Table 4 pharmaceutics-10-00263-t004:** FDA-approval drug products of injection delivery based on liposome technology (liposome, nanoparticle, nanosuspension, microemulsion, microsphere).

Drug Name	Active Ingredient	Composition/Type	Company	Indication	Approval Date
Liposome	-	-	-	-	-
New drugs	-	-	-	-	-
Doxil^®^	Doxorubicin hydrochloride	HSPC, cholesterol and PEG	Janssen	Ovarian Cancer; Sarcoma; Myeloma	1995
Ambisome^®^	Amphotericin B	HSPC, DSPG, cholesterol and amphotericin B	Astellas	Fungal infection	1997
Depocyt^®^	Cytarabine	Cholesterol, Triolein, DOPC and DPPG	Pacira	Lymphomatous	1999
Exparel^®^	Bupivacaine	DOPC and DOPE	Pacira	Local anesthetic	2011
Marqibo kit^®^	Vincrinstine Sulfate	Cholesterol and eggs sphingomyelin	Talon	Acute lymphoblastic leukemia	2012
Onivyde^®^	Irinotecan hydrochlorine	DSPC, MPEG-2000-DSPE	Ipsen	Adenocarcinoma of the pancreas	2015
Generic drugs	-	-	-	-	-
Doxorubicin hydrochloride	Doxorubicin hydrochloride	DSPC and cholesterol	Sun pharma	Ovarian cancer; sarcoma	2013
Doxorubicin hydrochloride	Doxorubicin hydrochloride	DSPC and cholesterol	Dr Reddys	Ovarian cancer; sarcoma	2017
Microsphere	-	-	-	-	-
Lupron Depot^®^	Leuprolide Acetate	PLGA	Abbvie	Advanced prostatic cancer	1989
Sandostatin Lar^®^	Octreotide acetate	PLGA	Novartis	Acromegaly	1998
Trelstar^®^	Triptorelin pamoate	PLGA	Allergen	Advanced prostate cancer	2000
Definity^®^	Perflutren	DPPA, DPPC and MPEG-5000-DPPE	Lantheus	Ultrasound contrast agent	2001
Risperdal Consta^®^	Risperidone	PLG	Janssen	Schizophrenia; Bipolar I Disorder	2003
Vivitrol^®^	Naltrexone	PLG	Alkermes	Alcohol dependence	2006
Bydureon^®^	Exenatide synthetic	PLGA	Astrazeneca AB	Type 2 diabetes	2012
Signifor Lar^®^	Pasireotide pamoate	PLGA	Novartis	Acromegaly	2014
Lumason^®^	Sulfur hexafluoride lipid-type microspheres	DSPC and DPPG-Na	Bracco	Ultrasound contrast agent	2014
Bydureon Bcise^®^	Exenatide	PLGA	Astrazeneca AB	Type 2 diabetes	2017
Triptodur Kit^®^	Triptorelin pamoate	PLGA	Arbor	Central precocious puberty	2017
Suspension and nanoparticle		-	-	-	-
Atridox^®^	Doxycycline hyclate	PLA	Tolmar	Chronic adult periodontitis	1998
Eligard^®^	Leuprolide acetate	PLGA(Atrigel^®^)	Tolmar	Advanced prostate cancer	2002
Abraxane^®^	Paclitaxel	Protein nanoparticle	Abraxis	Metastatic Breast Cancer; Non-Small Cell Lung Cancer	2005
Somatuline Depot^®^	Lanreotide acetate	Nanotube [74]	Ipsen	Acromegaly	2007
Zyprexa Relprevv^®^	Olanzapine pamoate	Microcrystal	Eli lilly	Schizophrenia	2009
Invega Sustenna^®^	Paliperidone palmitate	Nanocrystal	Janssen	Schizophrenia	2009
Feraheme^®^	Ferumoxytol	carbohydrate-coated iron-oxide nanoparticle	Amag	Iron deficiency anemia	2009
Sustol^®^	Granisetron	Ortho ester (Biochronomer™)	Heron	Nausea and vomiting	2012
Abilify Maintena^®^	Aripiprazole	Nanocrystal	Otsuka	Schizophrenia	2013
Ryanodex^®^	Dantrolene sodium	Nanocrystal	Eagle	Malignant hyperthermia	2014
Invega Trinza^®^	Paliperidone palmitate	Nanocrystal	Janssen	Schizophrenia	2015
Aristada^®^	Aripiprazole Lauroxil	Nanocrystal	Alkermes	Schizophrenia	2015
Sublocade^®^	Buprenorphine	PLGA	Indivior	Moderate to severe opioid use disorder	2017
Emulsion	-	-	-	-	-
Intralipid^®^	Soybean Oil	Fat Emulsion	Fresenius	Parenteral nutrition	1975
Cleviprex^®^	Clevidipine	Lipid emulsion	Chiesi	Reduction of blood pressure	2008
Perikabiven^®^	Amino acids	Lipid emulsion	Fresenius	Parenteral nutrition	2014
Smoflipid^®^	Fish oil	Lipid emulsion	Fresenius	Parenteral nutrition	2016
Cinvanti^®^	Aprepitant	Lipid emulsion	Heron	Acute and delayed nausea and vomiting	2017

DOPE, dioleoylphosphatidylethanolamine; DOPC, dioleoylphosphatidylcholine; DPPG, dipalmitoylphosphatidylglycerol; HSPC, hydrogenatedsoyphosphatidylcholine; DSPG, distearoylphosphatidylglycerol; DSPC, distearoylphosphatidylcholine; PEG 2000-DSPE, polyethylene glycol 2000-distearoylphosphatidylethanolamine; PLGA, PLG, poly(lactic-co-glycolic acid); PLA, Polylactic Acid.

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
