# Peer review of "A Comprehensive Map of FDA-Approved Pharmaceutical Products"

_pharmaceutics, 2018, doi:10.3390/pharmaceutics10040263_

Round 1

Reviewer 1 Report

The review by Hao Zhong et al is an interesting area where the information about all the FDA approved pharmaceutical products and their routes of administration are reviewed. The flow of the review is very well organised. The distribution of dosage forms, administration routes and clear picture of advanced pharmaceutical technologies including the recent advances were comprehensively described. Overall, the quality of the review was excellent and the manuscript can be accepted in its current form.

Author Response

Reply to the reviewers’ comments for Manuscript pharmaceutics-393790 (Title:A comprehensive map on FDA-approved pharmaceutical products)

First of all, we appreciate the editor and reviewers for their enlightening comments and suggestions on our manuscript. The manuscript has been revised according to the reviewers’ advises and the revised words were labeled with red and modified figures were uploaded again.

Here are the point-to-point answers to the editor and referees’ comments.

Reviewers' comments:

Reviewer 1:

1. The review by Hao Zhong et al is an interesting area where the information about all the FDA approved pharmaceutical products and their routes of administration are reviewed. The flow of the review is very well organized. The distribution of dosage forms, administration routes and clear picture of advanced pharmaceutical technologies including the recent advances were comprehensively described. Overall, the quality of the review was excellent and the manuscript can be accepted in its current form.

Answer: Thank you for the comments.

Reviewer 2 Report

This paper is a review of the success and failure of various FDA approved drug delivery instruments from clinical trials to their status in the market. Sufficient and appropriate literature have been cited.

However, the following comments will help improve the quality of the paper:

All claims made in the review, other than own data arising from synthesis of information, must be supported by relevant reference(s). Some claims which need referencing/ citation are: 

i. Lines 170 - 171

ii. Lines 196 - 197

iii. Lines 2523 - 253.

iv. Line 333.

Line 356 - It should be highlighted that though PGLA and PLA are synthetic polymers, they are both biodegradable and biocompatible.

English language should improved, with special attention to grammar and  to present a better and flow of information. This applies to the entire paper.

Author Response

Reply to the reviewers’ comments for Manuscript pharmaceutics-393790 (Title:A comprehensive map on FDA-approved pharmaceutical products)

First of all, we appreciate the editor and reviewers for their enlightening comments and suggestions on our manuscript. The manuscript has been revised according to the reviewers’ advices and the revised words were labeled with red and modified figures were uploaded again.

Here are the point-to-point answers to the editor and referees’ comments.

Reviewers' comments:

Reviewer 2:

1.     All claims made in the review, other than own data arising from synthesis of information, must be supported by relevant reference(s).

i. Lines 170 - 171

ii. Lines 196 - 197

iii. Lines 2523 - 253.

iv. Line 333.

Line 356 - It should be highlighted that though PGLA and PLA are synthetic polymers, they are both biodegradable and biocompatible.

Answer: Thank you for the suggestions. We have added the relevant references

to support these claims. For example:

1.     In lines 170-171, the patents protecting these novel drugs have been listed, such as < Non-mucoadhesive film dosage forms> and < Stabilized amine-containing actives in oral film compositions> for Zuplenz® oral soluble film. In addition, we have added some explanation on patent protections.

2.     In lines 196-197, < Formulation strategy and use of excipients in pulmonary drug delivery> was added.

3.     In lines 252-253, <Oral osmotically driven systems: 30 years of development and clinical use> was added

4.     In line 333, < Inorganic Nano-Targeted Drugs Delivery System and Its Application of Platinum-Based Anticancer Drugs> was added

5.     We have highlighted that though PGLA and PLA are synthetic polymers, they are both biodegradable and biocompatible in line 356.

2. English language should be improved, with special attention to grammar and  to present a better and flow of information. This applies to the entire paper.

Answer: Thank you for the suggestions. We have revised the grammar and did

the proof-reading to the manuscript.

Reviewer 3 Report

This review gives a very interesting update on drugs approved by FDA during the recent years. by the way, it would be also very interesting to prepare similar review for drugs approved in Europe and Asia. Some figures are too complicated (Fig. 3) or the colors used are very similar so it can be misleading for readers and therefore must be changed (Fig. 2). In table 5, it is strongly recommended to indicate which formulation represent which type nano-/microcarrier instead of a general statement in its caption. Please complete this review with the information on biopharmaceuticals as Fig. 1a shows a considerable growth in the number of these products on the US market. There is  statement on patent protection of some oral formulations in  Line 169. However, it is not clear what kind of novelty concerning swallowing difficulties is protected by the patent. This statement is too general, more details are needed. There are many references in the list which are not complete.

Author Response

Reply to the reviewers’ comments for Manuscript pharmaceutics-393790 (Title:A comprehensive map on FDA-approved pharmaceutical products)

First of all, we appreciate the editor and reviewers for their enlightening comments and suggestions on our manuscript. The manuscript has been revised according to the reviewers’ advices and the revised words were labeled with red and modified figures were uploaded again.

Here are the point-to-point answers to the editor and referees’ comments.

Reviewers’ comments:

Reviewer 3:

1. Some figures are too complicated (Fig. 3) or the colors used are very similar so it can be misleading for readers and therefore must be changed (Fig. 2).

Answer: Thank you for the suggestions. We have changed the colors in Fig. 2.

Fig. 3 is complicated, but this kind of figure can display a lot of data. For example,Fig.3 not only displays the distribution of dosage forms of all drugs, but also reveals the ratio of new and generic drugs.

2. In table 5, it is strongly recommended to indicate which formulation represent which type nano-/microcarrier instead of a general statement in its caption.

Answer: Thank you for the suggestions. We have added the related information

in table 5.

3. Please complete this review with the information on biopharmaceuticals as Fig. 1a shows a considerable growth in the number of these products on the US market.

Answer: Thank you for the suggestions. We have completed the review with

the information on biopharmaceuticals.

4. There is statement on patent protection of some oral formulations in Line 169. However, it is not clear what kind of novelty concerning swallowing difficulties is protected by the patent. This statement is too general, more details are needed.

Answer: Thank you for the suggestions. We have given an example to explain

on patent protection. The statement is following.

“For example, Zuplenz® is a unique formulation of ondansetron developed using PharmFilm® technology. This technology has been granted U.S patent in 2010, which are providing intellectual property protection for the Company's film products and methods of their preparation. And thus, Zuplenz® could enjoy a long-term market exclusivity.”

5. There are many references in the list which are not complete.

Answer: Thank you for the suggestions. We have Corrected these incomplete

References in the list.

Round 2

Reviewer 3 Report

I accept the corrections made in the manuscript but the references no 75, 78 & 80 are still incomplete as full bibliographic data is missing. It is very important for review articles as readers may want to find a source of information to analyze it in more depth. If full bibliographic data are not available (article in press) please mention clearly In press and provide doi number. Both the number of issue/volume and the number of pages should be given if appropriate (New York Times). The abbreviation of author's name is not always given.

Author Response

Reply to the reviewers’ comments for Manuscript pharmaceutics-393790 (Title: A comprehensive map on FDA-approved pharmaceutical products)

First of all, we appreciate the editor and reviewers for their enlightening comments and suggestions on our manuscript. The manuscript has been revised according to the reviewers’ advices and the revised words were labeled with red and modified figures were uploaded again.

Here are the point-to-point answers to the editor and referees’ comments.

Reviewers’ comments:

Reviewer 3:

1. I accept the corrections made in the manuscript but the references no 75, 78 & 80 are still incomplete as full bibliographic data is missing. It is very important for review articles as readers may want to find a source of information to analyze it in more depth. If full bibliographic data are not available (article in press) please mention clearly In press and provide doi number. Both the number of issue/volume and the number of pages should be given if appropriate (New York Times). The abbreviation of author's name is not always given.

Answer: Thank you for the suggestions. We have revised the references carefully.

1.     For article in press, we have mentioned clearly In press and provide doi number. “Yang, Y. and Z. Ye, Deep learning for in vitro prediction of pharmaceutical formulations. Acta Pharmaceutica Sinica B, 2018. https://doi.org/10.1016/j.apsb.2018.09.010 (In press).”

2.     For publication in New York Times, we have added relevant website link.

For example,

“Grady, D., FDA approves first gene-altering leukemia treatment, costing $475,000.NewYorkTimes.,2017.https://www.nytimes.com/2017/08/30/health/gene-therapycancer.html?mcubz=3.”

“Belluck, P., First Digital Pill Approved to Worries About Biomedical ‘Big Brother’.NewYorkTimes.,2017.https://www.nytimes.com/2017/11/13/health/digital-pill-fda.html.
